# Exploring the Representation of Cows on Dairy Product Packaging in Brazil and the United Kingdom

**Karynn Capilé** [1,*], **Claire Parkinson** [2], **Richard Twine** [2], **Erickson Leon Kovalski** [3] **and Rita Leal Paixão** [1]

1   Graduate Program in Bioethics, Applied Ethics and Public Health (PPGBIOS), Biomedical Institute,
    Fluminense Federal University, Rua Prof. Hernani Melo, 101-São Domingos, Niterói 24210-130, RJ, Brazil;
    ritapaixao@id.uff.br
2   Centre for Human Animal Studies (CfHAS), CE210 Creative Edge Building, Edge Hill University,
    St Helens Road, Ormskirk L39 4QP, UK; claire.parkinson@edgehill.ac.uk (C.P.); twiner@edgehill.ac.uk (R.T.)
3   Graduate Program in History of Science and Technics & Epistemology, Federal University of Rio de Janeiro
    Avenida Athos da Silveira Ramos, 274, NCE/CCMN-Cidade Universitária-Ilha do Fundão,
    Rio de Janeiro 21941-916, RJ, Brazil; erickson.leon@gmail.com
*   Correspondence: karynn.capile@gmail.com

**Abstract:** Food packages must communicate mandatory information, but they can also be used for marketing practices such as promotion and are a communication pathway from industry to consumer. Considering that cows are the main beings affected by the dairy industry, it is essential to scrutinise what dairy product packages convey about them. The aims of this study are to analyse the occurrence of reference to cows on the packaging of dairy products in popular supermarket retail stores in Brazil and the United Kingdom and to discuss ethical implications of promotional practices of dairy producers. We found that in both countries most packaging does not refer to cows at all. In the UK, an average of 31% of the packaging used some visual reference to cows, and in Brazil an average of 15% of packaging used some visual reference to cows. We identified four modalities of cow signifiers with a strong common appeal to nature that reflect and reaffirm an idyllic narrative of milk production. Our findings reflect the concept of absent referent, coined by Carol Adams, both on the packages containing some type of cow representation and on the packages not containing any. Considering that it might influence the consumer's understanding and attitude towards cows, we highlight that the lack of adequate information about cows' conditions and the obscuring of problematic issues in cows' exploitation through the globalization of the happy cow narrative are two important issues to be placed on the Marketing Ethics concerns.

**Keywords:** dairy industry; bioethics; happy cow; marketing; consumer awareness

## 1. Introduction

There is an ongoing debate about public awareness and attitudes regarding animal product consumption. For instance, citizen opinions on animal welfare issues differs widely amongst different countries and studied groups, indicating there is no common ground about the relevance of the conditions of life of animals raised for food in consumers' choice. Social psychologists have shown that people tend to evaluate animals' characteristics according to which purpose these animals are used for, regardless of their cognitive, emotional, or species-specifics capacities [1]. The emergence of the field of animal ethics in recent decades has increased scrutiny over the construction of meanings, moral values and choices that affect animals. Marketing is known to have strong influence on consumer behaviour [2], either for more sustainable choices or for the maintenance of pre-existing harmful practices [3].

We assume that if the marketing of animal products influences attitudes towards animal products consumption, identifying the messages that have been conveyed within different products is an important starting point of analysis to understand the complex

relation between meaning construction, decision-making and attitudes shift towards animals. Findings in this area can be useful for the planning of public policy or regulations to promote more sustainable attitudes.

Globally, milk and dairy have an important role in nutritional, cultural and economic basis of most nations, however, controversies regarding the dairy industry and reasons to replace its products by more sustainable options are increasing. In addition its economical contribution and importance for growth and development providing energy, protein, micronutrients and bioactive compounds [4], the environmental impact and consequences on societal well-being due to ethical concerns, are undeniably urgent and should be more discussed. Using biophysical models and methods and assuming corresponding changes in production, researchers estimated that replacing half of animal-derived food with plant-based foods on a daily basis would achieve a 40% reduction in nitrogen emission, 25–40% reduction in greenhouse gas emission and 23% per capita less use of cropland for food production [5]. Such changes in diet could also be beneficial from a health point of view, since fruits and vegetables intake are below recommended levels [6].

Although milk and dairy replacement, or at least reduction, could mitigate environmental issues, these reasons have been shown to be insufficient to promote substantial changes. Aiming to identify the messages about cows on milk and dairy products, this research explores what dairy marketing, in two different countries, may be communicating about cows to consumers through one of the main channels of communication between dairy companies and dairy consumers: the packages placed on food stores shelves. Apart from logistic and accessibility factors regarding the research groups involved, a comparison of cow representations in packages between Brazil and UK was proposed because we hypothesized that the differences in consumption rates, historical and cultural differences in terms of animal rights movements and socioeconomical identities could reflect different aspects of representations on packages. In addition, the findings could contribute to elucidate more efficient tactics to raise public awareness about the negative impacts of the dairy industry in different locations.

In the following section, before presenting our original results and analysis, we introduce the conceptual background of this research that includes some perspectives, theories and studies focused on the current understanding of the morality of human–animal relations, especially of consumption, major problems of cow's welfare, relevant aspects of packages as a promotional media and critical perspectives on animals products marketing.

## 2. Conceptual Background

### 2.1. Moral Conflicts and the Biased Endorsement of Animal Exploitation

Caring for animals but not changing attitudes in their favour has been addressed by many authors as a moral paradox in eating behaviour and human–animal relationships [1,7–12]. The ways people cope with the paradox or cognitive dissonance that results from the awareness of issues of animal exploitation might be mediated by multiple factors. These include individual moral backgrounds and personal beliefs, in addition to cultural habits that exploit other animals and are ideologically normalised within popular culture [11–15]. Ideology here refers to values and sets of beliefs and practices (intentional or not) that reflect it [16,17], and it is marked by four features: having cognitive power; having evaluative power; working as guidance towards action; and being logically coherent [18]. Media discourses and representations have an important association with ideology, not only reflecting but also reinforcing it [19–21]. For Adams this includes the mechanism of the 'absent referent' where meat and dairy production and marketing placates the consumer by hiding realist images of whole animals and removing the traces of violence inherent to their commodification [19]. Cole & Stewart [22], underlined the normalisation of milk consumption in childhood both at home and in schools, and how school food practices form part of the hidden curriculum that helps shape omnivorous eating as normative. Media discourses could be understood as contributing to such processes of normalization.

Changing eating practices may depend on ethical motivation but also on an enabling situation and the presence of relevant emotional conditions [12,23–25]. Despite rising criticism, using animals for food is a mainstream practice strongly established in the dietary cultures of most societies [26]. This subject has a parallel with a range of studies on social psychology about moral decision-making that involve settled meanings and social psychological mechanisms of selective deactivation and permissiveness that preventively dissociate current actions of a violent behaviour status [27]. Intentionally changing practices to benefit animals used for food and denaturalising the instrumental role assigned to them in human culture might be fostered by recognising them as sentient beings rather than just objects like food, food sources [28], commodities [29,30], or property [31].

The philosopher Elisa Aaltola highlights the relevance of the context and the cultural paradigm on human–animal relations. She evokes the ancient philosophical concept of *Akrasia* to refer to the paradoxes and incongruences between what people think they should do and what they really do towards animals [32]. Psychology scholars have drawn on the concept of ambivalence or moral disengagement [33,34] to explain such postures that some have called the *meat paradox*: a contradiction between affection, sympathy, and love for animals and the concomitant exploitation, operation, and use of them [7,10,16].

In addition, there is not a universally agreed concept of a good life for animals amongst consumers, rather their concerns about farm animals' conditions may arbitrarily vary depending on their personal values [35]. Therefore, when consumers declare that they care about animal welfare, it does not necessarily mean they have a grounded understanding of animal needs [36] or that they engage in more ethical choices [37].

Scholars have argued that people tend to justify the continuation of animal exploitation through rationalisations that strengthen their beliefs instead of looking for answers to resolve the cognitive dissonance generated by some undesirable information about the practice [33,34,38]. An opinion poll focused on meat eaters showed that most of them endorsed values that Joy identified as the basis of a carnist ideology [38]: consuming meat was claimed as normal, natural, and necessary—3N [16]. Further, since a significant number of people justified their consumption because it was pleasant, subsequent research began to consider the existence of one more N—nice—thus pointing to a 4N justifying speech [38]. Furthermore, investigations found that people denied that the animals they consume more often had internal lives, feelings, and moral value, indicating that classification criteria were not based on ethics, biology, or neuroscience but on the convenience of maintaining certain beliefs [1,10]. This corroborates the social intuitionist theory statement that moral judgments primarily start from intuitive ideas and later are rationalised, being strongly influenced by social shared beliefs that sustain common moral justifications. In this perspective, rationalisation is, in most cases, only a justification of pre-existing beliefs and values [39].

The moral disengagement theory supports that there is a selective moral disengagement in the exercise of moral agency [40]. According to this theory, a person's exercising of moral agency occurs through a self-regulatory process that could be selectively disabled to reduce dissonance and minimise the associated damage to his or her character; this deactivation is called moral disengagement [33,40]. Deactivation of moral evaluation mechanisms allows people to maintain harmful behaviour by blocking the growth of a negative self-image. So, even if certain conduct causes damage, it might not be seen as morally problematic. This theory has been used to address conflicting attitudes toward non-human animals, such as the disagreement to interrupt practices based on animal exploitation even in the face of a growing number of allegations against them [14]. Regarding animal products consumption, four patterns of deactivation have been described: (1) considering a harmful behaviour acceptable by highlighting the importance of ends or by using euphemistic labelling to minimise the damage associated with it, lining it up to moral standards, or by comparing the conduct with a worse one; (2) denying the responsibility and the individual's importance over the damage by projecting the responsibility on to a larger group or describing the conduct as a result of external pressures; (3) misrepresentation of

consequences, selective unawareness, denying cause–effect relations, or minimising negative consequences; (4) dehumanising the harmed individuals by denying them sentience, cognitive capacities, and moral relevance [33]. Therefore, psychological mechanisms to avoid internal moral conflict combined with a powerful invisible ideology that is constantly reinforced and encouraged through popular media, such as animal product marketing, might generate a bias in society's understanding and its attitude towards animals [16,41,42].

The more animal ethics debates are seen as relevant in global spheres, such as global ethics, global justice [43,44], global health [45], cosmopolitan rights [46,47], and politics [48–51], the more veganism is claimed as a reasonable political project [52–54]. This constitutes a source of cognitive dissonance for animal product consumers [55,56]. In contrast, global food culture and marketing still stimulate animal product consumption. Thus, figuring out how to effectively encourage changing eating habits that are socially normalised for ethical and political reasons is still an ultimate challenge [57,58] hampered by the corporative interests in maintaining animal exploitation and promoting animal product consumption, in addition to the consumer's moral mechanisms that support harmful food practices [14].

### 2.2. Cows' Exploitation Issues

We summarise the criticism of the exploitation of animals for food along three main axes: ethics regarding animals' interests; human health issues; and environmental damage. In this article, we draw more attention to the ethical issues of animal exploitation which manifests itself as a sufficient reason for addressing the subject matter. The impacts on human health, on the other hand, heighten the problem represented in the first axis, since the replacement of animal products in the human diet does not imply health loss and may even result in health improvement [59–61]. Whereas the environmental damage amplifies the problem even more due to its negative impacts on human and non-human quality of life, especially if taking the future generations into account. Indeed, the long-term harm poses an existential risk not only for humans but for biodiversity in general. There are a growing number of studies that identify factory farming as a significant cause of environmental damage [62–64], studies that recommend replacement or reduction of animal product consumption as a better choice for both health promotion [65–68] and environmental damage mitigation [5,57,67,69,70], and studies on the ethical issues of using animals for food [71–75], in addition to documentaries, books, papers, and activist discourses on popular media that have put this subject on the political agenda [76].

Dairy farming is increasingly associated with environmental harm [77–79] and animal welfare problems [80]. Highly common causes of welfare decline in dairy farming are mastitis [81–84], zero grazing (no access to pasture) [85], invasive procedures without anaesthesia and pain management such as dehorning [86], hock lesions [83], ketosis and gait abnormalities [87], lameness and foot disorders [88], heat stress, metritis and other reproductive disorders, respiratory diseases, social stress and abnormal behaviours like fights and stereotypies [80]. In addition to all these cow welfare issues in dairy farming, there is still cow–calf separation, a highly stressful event for both the cow and calf. Early separation from the newly born calf is a major subject in cow welfare about which many dairy consumers are unaware [15,89,90].

Although the relation between naturalness and welfare is not causal, the intensification of production, which implies a loss of naturalness, tends to also trigger welfare impoverishing. In the industrial farming factory cows are genetically selected and handled to produce more in less time and less space. Subsequently, the animals have no chance to express many of their natural behaviours. These circumstances lead to a decline of the welfare level [82,91]. In the United States, for instance, the rate of production has increased about three to four times compared to the rates of 60 years ago [92].

Another problem is the desensitisation of cow handlers to unhealthy conditions and behaviours that tend to be seen as normal due to the high percentage of occurrences. As many cows have some level of mastitis and lameness, for instance, these become acceptable

conditions regardless of consequences to cow welfare, such as chronic pain [85]. Whereas cows' health and welfare have an instrumental economic value for the dairy industry [35], cows' issues that do not directly affect economic gain will likely remain unresolved.

The negative aspects of dairy farming mentioned above characterise a vulnerable condition of dairy cows and pose a major concern to sentient-centric ethical perspectives, in which sentient beings matter regardless of their instrumental or relative value [31,48,93–97]. Apart from the actual direct impact of exploitation on cows' welfare, the lack of a cow's intrinsic value is a permanent welfare issue for the dairy industry, in which welfare needs are not a goal per se. An example of how cows' interests might be dismissed is grazing management. In terms of cows' behavioural needs, the better practice is a total grazing system [91], which matches with public preference [81]. However, a total grazing system is not economically advantageous compared to the combination of grazing and industrial systems, referred to as a mixed system, which is considered a better option in terms of sustainability [98].

Despite the evidence that declining animal consumption would be a positive social achievement, world demand for meat, dairy, fish products, and animal proteins is expected to increase two-fold by 2050 [99,100]. In the same direction, milk consumption is increasing around the globe [57,77,101], in tandem with the intensification of milk production and consequent negative environmental impacts of dairy farming [79,102,103], even in mountainous regions that are traditionally known for low-intensive farming systems [104]. The largest milk consumption in the world is in developed countries, and it is anticipated that developing countries will reach similarly high rates in the future [101]. According to a study from 2009 on the impact of diet on climate change, the group of countries that compose the Organisation for Economic Co-operation and Development (OECD) (in which the UK is included) has the highest rates of milk consumption (kg/per capita/year), followed by Brazil, Russia, India, and China (BRIC) [105]. It is expected that worldwide consumption will increase absolutely and per capita and may increase by more than 50% from 2010 to 2050 [105,106]. If reducing consumption is the most effective solution to mitigate damages [67,68] in all axes, it is urgent to develop strategies to prepare society to make this change. There is a growing alertness concerning the bucolic idea of dairy farming that the industry sustains [55,107–109] due to the fact that it is not a reliable depiction of the cow's circumstance. A 'flagship' for dairy marketing is the *California happy cow* campaign, which debuted in 2007 in the United States, that heavily invested in the happy cow narrative [110,111]. The animal rights organisation People for the Ethical Treatment of Animals (PETA) sued the California Milk Advisory Board (CMAB) more than once, in vain, for deception [112]. Moreover, many scholars and activists have argued that the message conveyed by this campaign is problematic and misleading [85,92,109,112,113]. However, the advertisement was never banned. The California happy cow became an emblematic instance of the widespread untruthful portrayal of dairy farming and of the economic power relations behind this industry.

More recently, a pro-vegan organisation operating in the UK designed a campaign that used the slogan *milk is inhumane* and emphasised the welfare issues that result from regular dairy farming practices, especially cow–calf separation. People from the dairy industry complained to the Advertisement Authority Association (ASA) accusing the *Go Vegan* campaign of misleading advertisement, but the ASA dismissed the dairy farmers' case [114]. This is a wholly different outcome to the case against the California happy cow in the United States, and it may reflect an interesting change of course.

### 2.3. Towards an Expanded Concept of Sustainability

The Food and Agriculture Organization of the United Nations (FAO), one of the main worldwide sources of information, guidelines and references on sustainable agricultural practices, states that animal farming is detrimental to the environment, public health and animal welfare in many respects. In FAO's Guidelines for Sustainability Assessment of Food and Agriculture Systems (SAFA), animal welfare is included in environmental

integrity and they emphasize that 'ethical considerations are a major reason to take care of animal welfare' [115].

Nevertheless, even the explicit allusions of animal welfare in sustainability definitions, principles and assessment criteria do not necessarily imply a direct regard or intrinsic value on the animals' needs and quality of life. What is commonly observed is a human-centric discourse linking sustainability and animal welfare [116]. Similarly, the concept 'one welfare' (derived from the concept 'one health') suggests that the improvement of human and animal welfare must be both pursued and equally valued. However the declared reason for this pursuit is hardly grounded on the basis that animals are important per se. The recurrent premise is that animal health is essential for human health, as well as animal welfare is a condition to reach and safeguard human welfare [117].

On the other hand, the sustainability sphere has evolved since its initial alignments, at least in conceptual terms, and there have been efforts to include the sense of intrinsic value of animals in the conceptual scope and practices to raise more sustainable attitudes and institutions. Probyn-Rapsey et al. [118] defended an expanded version of 'sustainability framework' that takes into consideration the issue of the carbon emission from animal agriculture and the inclusion of a non-anthropocentric ethical perspective as an integral part of social justice. In this sense an institution that aspires to be sustainable needs necessarily to consider the substitution of animal-based food for options more aligned with social justice purposes. This categorisation shift from environmental to social is an update on the FAO's concept that brackets animals with natural resources such as water, land, atmosphere, biodiversity, materials, and energy.

### 2.4. Packages as Promotional Media

Packaging has mandatory functions, which, in terms of food products, means containing, protecting, preserving, and informing, but it might also be used for promotion [119,120]. The physical structure and material are also important aspects. Amongst the main types of textual messages in packages are the brand name, secondary copy (short description of the product) and romance copy (claims or pictures emphasizing the quality of the product that are not mandated by regulations). Labelling exigencies include the compulsory details of the mandatory copy, such as nutritional facts and ingredients, weight, and percentage of components according to different classes of products. The remaining space on the package surface can be used for promotion, involving, in this case, the same resources and patterns of printed media as colour, text, typography, imagery, icons, and symbols [120]. Just as in printed media, visual communication involves verbal and visual codes that interact with each other [121]. In this study, we address specifically the non-mandatory elements of the textual messages of packages and focus on the graphic design, which is mainly related to the promotional marketing.

Although there are legal requirements to reserve some space for nutritional facts, contents, and warnings about allergenic components, which is impartial information [120], there are no strict rules for the promotional messages on food packages [122]. In the UK and Brazil, as in the United States and many other countries, advertising is subject to a self-regulated code, but there is no specific institution to oversee the regulation of promotional aspects of animal product packages. Hence, the importance of the promotional role of packages on consumers' behaviour and attitude has been underappreciated.

Visual communication is a powerful way to transmit many messages with explicit or implicit meanings that might affect social behaviour [123]. Images of animals are ubiquitous in visual media and widely considered effective in promoting products, creating brand identities or encouraging attachment [124].

In terms of food safety and public health, there is much debate over the need to warn consumers about risks and make efforts to clarify the health implications of consuming some products, such as by highlighting high rates of sugar or fat [45,125]. Even in cases in which health issues related to the consumption or use of some components are consensual amongst health institutions, promotional practices do not change as a result, and it is

often necessary to state strict rules or even bans to put pressure on marketing practices to meet global health demands, especially if the change might threaten profit interests [125]. The case of tobacco is a remarkable example of a concerted campaign in many countries motivated by the attempt to decrease health issues in which marketing practices were obligated to collaborate by stopping promotional messages on packaging and warning consumers about the potential harm of the product [126–129]. This illustrates the constant conflict of interest between corporate benefits and ethical principles.

### 2.5. Animal Product Marketing under Critical Perspectives

Critical media studies and critical animal studies problematize the exploitation of non-human animals and the key role of the media in the reinforcement and perpetuation of this exploitation [42,58]. They also advocate for a democratic media and criticise the dominant commercial communication that normalises exploitation of non-human animals and relationships of power [42,50,130,131]. Critical media and animal studies address marginalised themes and a lack of criticism, and they call for deeper understanding to scrutinise ideologies and cultural patterns that maintain social inequities. They coalesce around a critical interrogation of the 'animal-industrial complex' [132,133], defined 'as a partly opaque and multiple set of networks and relationships between the corporate (agricultural) sector, governments, and public and private science. With economic, cultural, social and affective dimensions it encompasses an extensive range of practices, technologies, images, identities and markets' [132]. In relation to dairy products critical analyses include Stewart and Cole's [22] examination of infant formula labelling, and Cole's [134] reading of milk advertiser's attempts to associate milk drinking with a male urban hipster identity. Such examples underline an animal-industrial complex reliant upon obfuscation and opacity.

From a different direction, but with a similar ethical scrutiny, a bioethical approach aims to identify conflicts of interest and those mainly affected by them, especially the vulnerable ones, and it seeks a resolution. Due to the narrowing of bioethics to medical themes in past decades, the concept of critical bioethics has been proposed to emphasise meticulous, critical, reflexive, and interdisciplinary investigation of a wide scope—with non-human animals included as morally relevant individuals—and problematization of relationships of power and social inequalities [50,135,136].

When questioned about their corporate social responsibility (CSR), manufacturers of products highly associated with health problems, such as tobacco and sugar-sweetened beverages, tend to focus on consumers' information and freedom of choice rather than on industry practices [125]. The assumption that the average adult consumer is critical and reflexive and makes deliberate choices is strongly contested by scholars of social psychology, behavioural economics, decision-making studies, and psychology of judgment [23,137–141].

Regarding animal product promotion, consumers may be easily deceived, not only because of the massive and aggressive marketing practices but also because of the lack of knowledge about farm animals' conditions [42,134,142], which need to be taken into account in marketing and communication due to their fundamental role in influencing consumer behaviour [143–145]. Discursive strategies, messages and meanings conveyed by the commercial media may influence a society's view and understanding about a given subject [146]. Therefore, the message that milk/dairy product packages communicate about cows matters.

### 2.6. Characterization of the Nations Involved in the Research

Brazil is a developing country with an economy strongly based on producing and exporting agricultural commodities, while also being the largest South American country. It holds the biggest commercial herd in the world [147] with an estimated herd size of beef and dairy cattle of 214.8 million heads in 2017 [148]. The milk production of 33,490 million litres accounts for the world's fifth largest milk producer in 2017 [149] and the per capita consumption/year was approximately 162 litres (Figure 1) [148,150,151].

Since it is associated with a higher income, the UK, also smaller geographically, has higher consumption rates [152]. In the UK, the estimated herd size of beef and dairy cattle in 2017 was 9.8 million heads [153], while the milk production was 14,708 million litres and per capita consumption/year was approximately 225 litres [154]. The UK was the 10th largest milk producer in the world in 2016 [155].

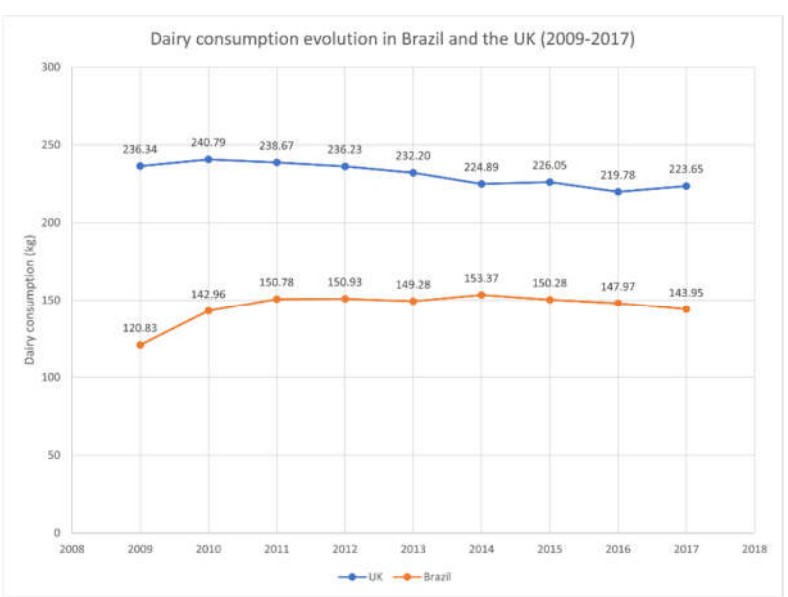

**Figure 1.** Evolution of average supply of milk across the population, measured in kilograms per person per year in Brazil (butter is excluded). Data from FAOSTAT [156] and Our World in Data [157].

## 3. Methods

### 3.1. Preliminary Phase

At an initial stage, during the first half of 2017, we visited a range of popular retail stores in both the UK and Brazil, taking pictures of animal representations from diverse categories of animal products. We analysed the imagery obtained identifying the products that had more animal representations and different types of representations (see imagery in Supplementary Materials). From this initial survey of the diverse range of animal representations on food packaging it was decided to focus the study on dairy product packaging for two reasons. First, this allowed an examination of the intertextual relationships between cow imagery across a range of different products that included milk, yoghurt, butter, and cheese. Second, there is less public awareness of ethical concerns about dairy production compared to meat and egg production, both of which have been the focus of high-profile public animal welfare campaigns. The success of such campaigns is reflected in the number of people that exclude milk and dairy from their diets which is much lower than those that exclude meat [158]. In this regard, it is notable that many vegetarians who express ethical concerns towards animals continue to consume dairy products. We were therefore interested to examine how the absent referent functioned in relation to animal product packaging where there was notably less public awareness of ethical issues relating to production. Therefore, from the initial findings regarding animal representations on animal product packaging, we narrowed down the scope of the study to dairy product packaging. We chose the visual representation of cows as the main identification criterion, as there was no evidence of linguistic references to cows without accompanying visual representations. We identified the repeated patterns of visual representations of cows and classified them based on the concept of modality. From semiotics, modality refers to the reliability and realism of the representation and the way the information is encoded in the message [17,120,146,159]. Based on these exploratory data, for the next phase we decided to focus on three retail chains in each country.

### 3.2. Case Selection

Three chains from each country were chosen due to their wide distribution throughout the national territory and variety of products (from several regions of the country, reflecting a good national sample) that target different classes. In Brazil, data was collected in Rio de Janeiro, the second most populous city of the country, from three leading supermarkets: Extra—from the Companhia Brasileira de Distribuição, Carrefour, and Prezunic—from the Cencosud company. The chain Zona Sul or the Pão de Açúcar stores focus on upper economic classes and display less variety and were thus not included in the study. In addition, these are not well spread over the cities: Zona Sul is only present in the south and central area of Rio de Janeiro, while Extra, Carrefour and Prezunic are widely spread and target middle and lower classes. Similar counterparts were selected in the UK: Tesco, Asda and Aldi also target middle and lower classes, offer a big variety and are well spread even in marginal areas over the cities, while Sainsbury's chain, on the other hand, has smaller stores, less variety of products and higher prices. We visited the biggest store of each of the three chains in the Liverpool City region, the fifth largest metropolitan area in the UK.

### 3.3. Data Collection

The sampling procedure consisted of taking photos of the products on the shelves in each visited store (see imagery in Supplementary Materials). We bracketed product packages in the following categories based on their customary aisles in stores: milk packages, yoghurt packages, ice cream packages, flavoured milk packages, cheese packages, and butter packages. The first one includes any cow's milk in liquid form, including fresh, long-life/sterilised, organic, and free-range milk. The second category includes yoghurt, fromage frais, and fermented milk. The third includes milk ice cream, lollies, and frozen yoghurt. The fourth category includes flavoured milk and milk shakes. The fifth includes any package containing only cheese, from soft to hard varieties, and cheese in bar, sliced, grated, or cream/spreadable forms, but excluding packages with any other content besides cheese. The last category includes packages of cow's milk butter, salted or unsalted, excluding mixed butter or margarine. We excluded other milk-based products due to the lack of consistency regarding the classification or composition, like those labelled as desserts, for instance, in which the concentration of ingredients may vary. We also excluded products with highly variable availability. During the data collection, we quantified the existing package designs in each of the six visited supermarkets, and then we calculated the proportion of packages that somehow referred to cows using visual messages on any of the package panel sides. In addition to representations of cows, we looked for verbal messages about grazing and milking systems. Data collection occurred between 2017 and 2018.

### 3.4. Imagery Analysis

To analyse the package images collected, we firstly classified them according to three categories of signs: icon, index, and symbol [17]. The iconic sign has physical resemblance to the signified; the meaning is intuitive. The indexical sign has nuanced evidence about what is being signified; the meaning may be inferred. The symbolic icon has no resemblance between signifier and signified; the meaning must be learned [17,146]. Then we distinguished and named modalities from representation that we found in the preliminary survey. The iconic signs were classified according to four modalities of representation: (1) Cartoon cows in a funny, playful, or comic atmosphere, caricatured, in sketch or childish drawing style, fun appeal [122]. (2) Pastoral cows grazing freely or being manually milked in a pleasant green field or old-fashioned farm, resembling pastoral painterly traditions. The main feature of this modality of representation is relating dairy products to tradition, nature, and health, in opposition to industrialised and intense production [109]. (3) Instagram-like photo: front view, highly colourful landscape in the background. Applies filters and highlights certain details of a real image. (4) Realistic photographic ownership portrait: farmers proudly posing next to or handling cows (Figure 2). Written text on the packaging

was included in the analysis where it was an integral aspect of the cow imagery (e.g., as a speech bubble). Other written text on the packaging was not included in the analysis.

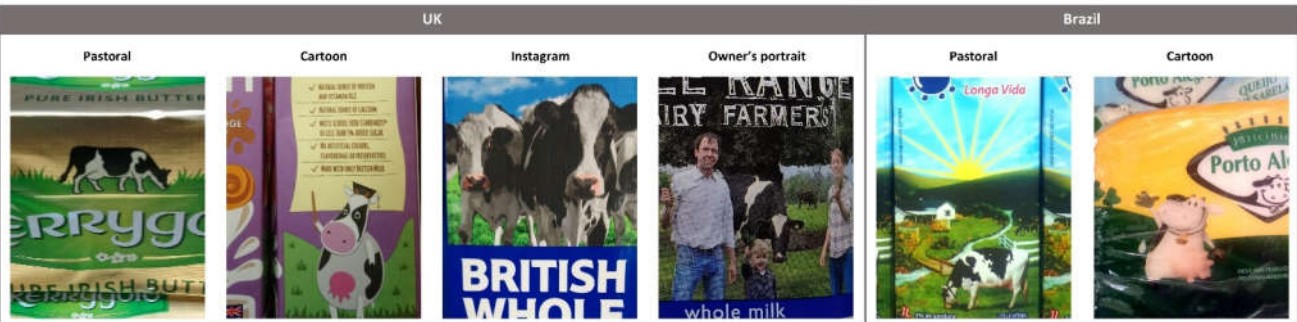

**Figure 2.** Examples of images from each modality of cow reference by country.

When an image presented elements of more than one modality, the sample was allocated in accordance with the predominant type of visual cues (Table 1). In addition to the package design, we also consulted other textual information about promotion strategies, profile, and brand identity on the websites of the respective brands. Regarding the graphic design patterns, we only counted each design once.

**Table 1.** Modalities of representations and the visual cues that characterise them.

| Modality | Visual Cues |
| --- | --- |
| Pastoral | Natural behaviours (grazing, positive social interaction, standing/laying relaxed, resting), lateral standing, natural elements (flowers, mountain, lake, animals, trees), old-style farm objects (milk bucket, wood fence, cow bell collar). Pastoral painterly tradition aesthetics. |
| Cartoon | Anthropomorphised behaviour/expressions, feminised features, forward-facing eyes, speech bubble, cuteness, comic/humour, targeted to children, fun names, unusual shapes and tastes, colourful scenarios, interactive elements. |
| Instagram | Realistic photographic image filling the front panel, close view of the cow, a few elements beside the cow(s), blue sky in the background, cow(s) looking to the camera (consumer). Positive behaviour reference (curiosity, closeness). Image filters. |
| Owner's portrait | People looking like the owner or carrier (vet, farm worker) standing close to the cow(s), embracing or feeding them, expressing pride/satisfaction, traditional family references, positive human–animal relationship/handling. |

## 4. Results

Regarding the occurrence of dairy packages referring to cows in the visited stores, we found that from the total of packages in the UK stores, 31% (37 of 120) of packages from Asda, 32% (36 of 113) from Tesco, and 31% (12 of 39) from Aldi had some cow representation. In Brazil, 16% (15 of 96) from Carrefour, 14% (13 of 91) from Extra, and 14% (12 of 87) from Prezunic had some cow representation. In both countries, amongst dairy products, milk packages most often displayed the cow sign, which corresponded to 67% of milk packages at Asda, 83% at Tesco, and 75% at Aldi. In Brazil, 31% of milk packages at Carrefour, 18% at Extra, and 36% at Prezunic displayed the cow sign (Table 2).

**Table 2.** Total distribution of dairy packages referring to cows in the visited stores.

|           | AS [1]     | TE         | AL        | CA [2]    | EX        | PR        |
|-----------|-----------|------------|-----------|-----------|-----------|-----------|
| Milk      | 67% (12)  | 83% (6)    | 75% (4)   | 31% (13)  | 18% (11)  | 36% (14)  |
| Flavoured | 40% (5)   | 80% (5)    | 67% (3)   | 11% (9)   | 14% (7)   | 0% (6)    |
| Butter    | 38% (8)   | 17% (6)    | 0% (2)    | 20% (5)   | 25% (8)   | 25% (8)   |
| Cheese    | 24% (37)  | 28% (36)   | 43% (14)  | 24% (25)  | 19% (26)  | 13% (24)  |
| Yoghurt   | 30% (30)  | 28% (25)   | 0% (8)    | 11% (27)  | 9% (23%)  | 8% (24)   |
| Ice cream | 21% (28)  | 26% (35)   | 13% (8)   | 0% (17)   | 6% (16)   | 0% (11)   |
| Total     | 31% (120) | 32% (113)  | 31% (39)  | 16% (96)  | 14% 91)   | 14% (87)  |

[1] UK stores: AS: Asda; TE: Tesco; AL: Aldi. [2] Brazilian stores: CA: Carrefour; EX: Extra; PR: Prezunic.

In respect to other dairy products, results refer to the occurrence of packages referring to cows on the shelves in different sectors, so if a brand had the same graphic design for various products, such as butter and cheese, it was counted as different items (even when the design was the same) because it was in different places (Table 3).

**Table 3.** Graphic designs found in each store.

|                  | AS [1] (37) | TE (36) | AL (12) | CA [2] (15) | EX (13) | PR (12) |
|------------------|-------------|---------|---------|-------------|---------|---------|
| Pastoral         | 15          | 7       | 3       | 8           | 8       | 7       |
| Cartoon          | 17          | 27      | 8       | 6           | 7       | 3       |
| Instagram        | 2           | 1       | 1       | 0           | 0       | 0       |
| Owner's portrait | 2           | 0       | 0       | 0           | 0       | 0       |

[1] UK stores: AS: Asda; TE: Tesco; AL: Aldi. [2] Brazilian stores: CA: Carrefour; EX: Extra; PR: Prezunic.

In the UK, 56 instances of cow references were found, with one indexical exception, of which 30 were from the cartoon modality, 18 from the pastoral modality, five from the Instagram modality, and two from the owner's portrait modality. In Brazil, from a total of 24 graphic design patterns of packages referring to cows, we found three indexical signs: a representation of a cow's udder under the label 'integral' on the milk package, another was a representation of cow spots along the milk package, and the other was an ice cream package displaying in the corner of the front side of the package a representation of an old-style milk bucket stamped with cow spots. Amongst the iconic representations, 13 were of the pastoral modality and seven were a cartoon. No instances of the Instagram or owner's portrait modality were found. Pastoral was the main modality in Brazil, and cartoon was the main modality in the UK.

From the 24 patterns of cow representations found in stores visited in Brazil, including the indexical and regardless of the modality, 21 (87%) were spotted cows, mainly black and white, only one was brown and white, and some were of unclassified colour. From the pastoral category, nine patterns were cows grazing—with the head pointing to the ground toward the grass—and four were just standing over the grass. The reference to nature and tradition was prevalent in all modalities of samples from both countries. All cartoon and pastoral modalities had a range of references to naturalness, such as flowers, mountains, ponds, and other animals like birds or insects in harmonic co-existence.

On most of the packages with cow representations, the verbal message was not referring directly to cows or to the farming features. In the UK sample, some of the cow representations were speaking to the consumer through a speech bubble; this was not observed in the Brazilian sample.

Dairy product packages in the popular supermarkets in Brazil and the UK did not communicate relevant information about cows, such as the grazing system (zero grazing, freely grazing, mixed grazing/housing), milking system (automatically, mechanically, or manually milking), or cow–calf separation system. Conversely, they did communicate misrepresented versions of dairy cows' lives, which we explore further below.

## 5. Discussion

### 5.1. Dairy Marketing Narratives

Our findings, in accord with similar investigations [107,160,161], point out that there is a global unchallenged ideology/narrative present in everyday life about milk production that the dairy industry cashes in on to promote its products. This is reinforced by visual messages that the milk comes from happy cows raised in a pleasant environment, traditional family farms, and that it is natural and healthy to consume it [107,131]. Visual messages are even more powerful than verbal ones due to the naturalisation of the connection between products and the images they represent; the audience is led to take those connections for granted and bypass their inherent contradictory condition, and visual syntaxes can convey messages that could not be verbally expressed [17,121,123,159,162–164].

### 5.2. The Absence of Cows

According to the concept of absent referent [19], once animals are made into food, they become an absent referent in the food discourse. The idea of a sentient being prior to the meat is totally eliminated from the food scenario. Despite the fact that meat comes from a sentient being, words and images of meat do not refer to them as subjects at all, but pieces of edible objects. Fragmenting, objectifying, and renaming pieces of animal cadavers as beef, hamburger, sausage, rib, nuggets, pepperoni, ham, or barbecue, for example, are part of the process of becoming [19]. In meat marketing, these elements are particularly evident, as the real animal is avoided and replaced by iconic images that mask the sentience of these beings, disguising their subjectivity. According to a survey, how we present, prepare, and talk about meat increases the willingness to eat meat by reducing empathy and disgust [13], corroborating Carol Adams's [19] argument that the way we refer to animals is related to our attitudes towards them. We point out that the cow is also an absent referent in the dairy marketing discourse, in two different ways, based on our findings. The more obvious way is the absence of any kind of reference to cows on most packages (around 85% of the packages from Brazil and 70% of the packages from UK), and the more subtle way is that when there is some representation of cow, it usually did not correspond to a real dairy cow, but to a fictitious character, thus, in both cases the cow's subjectivity is hidden even on packages displaying some cow representation. We suggest that this is an absent referent instance in the sense that the cow sign does not represent the subjective cow, an emotionally complex individual with interests and preferences that is kept behind the milk production discourse.

Using animal representations is a common and widely accepted element in promotional marketing in general [124]. One of the reasons it is so valuable for communication is precisely its metaphoric power. However, when the animals displayed on packages are the ones whose bodies the product comes from, they become metaphors of themselves. If cows on packages are supposed to refer to the ones the milk comes from, the message is unreliable. However, if they are not supposed to be representations of cows on real dairy farms, what is the point of using cow narratives rather than generic mascots if cow narratives might cause ambiguity in consumers' understanding? This is an important question, especially because consumers might be easily unaware about messages received from packages. In addition, a distorted cow narrative might overlap the actual dairy cows' conditions, which remain unknown for citizens due to the unfamiliarity with the current mechanized systems and little access to dairy industry facilities, process, and routine [165]. In a survey conducted among Brazilian citizens, for instance, most respondents were uninformed that cows generally do not have free access to pasture and are separated from their calf at birth [166].

### 5.3. The Meaning of the Cows on Dairy Packages

According to previous studies, cows appear in milk advertisements as instruments of production [167]; our findings corroborate this, especially in the cartoon and pastoral categories. In the UK the cartoon modality is predominant (Table 3) and most of the

package designs are explicitly targeting children, like those displaying the word 'kids' on the label, or implicitly targeting them, like those displaying childish images. This may result from the fact that the Dairy UK—organization that represents dairy producers—recommends the consumption of cow's milk from the age of one year and up [168]. In face of declining sales [169], the approach of milk as healthy and beneficial for young children is key to maintain a strong consumer market for cow's milk. Stewart & Cole [22] criticized the use of animal cartoon imagery which they found in their analysis of milk formula products. Constructing cuteness via cartoon representations could be seen as particularly effective in obfuscating the absented realities of the commodified animal. On the other hand, the predominance of the pastoral modality in Brazil (Table 3) may be related to the historically agricultural identity that has been emphasized in the last few decades with the significant investments and the growth of the agribusiness [170,171], along with government's promotion and encouragement of this sector [172,173]. The importance of the rural lifestyle in the Brazilian contemporary culture and popular aesthetics is on the rise even in major urban centres through the entertainment, music, fashion, food and sports industries [171,174].

The photographic-realistic representations as in the Instagram-like and the owner's portrait modalities found only in the sample from the UK have more realistic traits than idyllic ones. In print advertisement, using a photograph is a way to produce consciousness of realism [146], which may be a response from the marketers to the consumers' demands for transparency and origin assurance. This still maintains the happy cow narrative but through another meaning path. Like the pastoral, the photographic realistic representation also conveys naturalness, and this might meet consumers' expectations based on their bias that naturalness implies better welfare. It is not clear if using more photographic representations to convey realism is a new trend or if it is a specific feature of the marketing for organic and free-range products that claim to be more sustainable. In the sample obtained from the UK stores, three instances of the Instagram-like modality were found: two of regular British milk and one of cheese; none of them had an organic or free-range label or allusions to it. Perhaps another purpose of this type of representation is to convey transparency, which is also reported as highly valued by consumers [175], in response to the criticism of the fictitious and deceptive typical landscape of the dairy fairy tale in dairy marketing.

We found only two instances of the owner's portrait modality, both in the UK; one had the free-range title on the front of the package, and the other was labelled as organic. The controversial green marketing and its opportunistic appeals to sustainability and concern for negative impacts on 'livestock' have been largely criticised [108,176]. The owner's portrait modality of reference might be in consonance with a trend in dairy farming that compares cows to workers, arguing they express a collaborative behaviour that indicates a type of engagement or interest in participating in this industry [177]. Reference to the harmonic interaction is also evident in the representation of cows as they are milked by hand on both the Brazilian and UK packages: the cow is shown smiling and expressing joy in the situation. This may indicate how media and language do not create meanings but they are all together integrated in a bigger complex of meanings in which consumers are not just receivers but active participants [109]. The discourse of cows as workers [177] who desire to be milked [178] is perhaps an extreme variant of the happy cow narrative. This idea fits in one of the categories of selective deactivation of mechanisms of self-moral regulation [33] in two aspects. One uses the euphemistic labelling [33] 'worker' to minimise the harm associated with the meaning of 'exploited cow' and then aligns it with moral standards; the other advantageously compares [14] a conduct to a worse one by arguing that cows would be in worse conditions if they were not 'working', and thus this collaborative relationship is advantageous for them too.

In the cartoon modality, humour is one of the main cues. According to some authors, humour might be used as a strategy of keeping deceptive claims unnoticed [179]. What constitutes a deceptive claim might be considered a subjective question, resulting in such

occurrence not being problematized at all. However, it is not only outright lies that should be criticised; there are different types and degrees of deception. According to previous classification, the main types of deception are (1) vague/ambiguous: a phrase that is too broad or a statement with no clear meaning; (2) omission: omits information necessary to evaluate truthfulness or reasonableness; and (3) a false/outright lie: inaccurate or fabricated claim [179]. In general, deception is almost inherent to advertisement communication as it uses metaphors, humour, and subtle messages all the time; therefore, criticising the existence of deception in this media could result in empty criticism. Accordingly, what shall be prioritised in the ethical debates is the seriousness of certain deceptions, not the mere presence of metaphors or tricky language [179]. In this sense, the deceptive reference to cows is an issue because it might hinder the social recognition of cows as subjects and make it difficult to change consumers' attitude and behaviour toward them.

If parents' choice in the store is based on their children's preferences [164,180], then cartoons may be a promotional strategy that masks a controversial theme and helps to decrease cognitive dissonance towards cows while appealing to children. Many brands of dairy products, especially in the UK, have a range of products for children, regularly displaying cartoonish cows on the front of the package. This is even more problematic since children due to a lack of critical understanding, tend to believe more in advertising messages [181]. Furthermore, the cultural references regarding animals on literature, films and animal products marketing that target children contribute to a conceptual disassociation of the animals they are encouraged to consume from the animals they care about. This cultural learning reinforces the arbitrary attribution of social roles and moral status to animals according to their uses for human society [182]. Consequently, the place of a cow in society for consumers' and citizens' understanding might be carelessly established, with no assurance of further occasions to rethink it, particularly considering that dairy marketing worldwide will discourage critical thinking about dairy cows' exploitation. The representation of a bucket was found on many packages in the cartoonish and pastoral modalities; however, in the Western world, and even more in developed countries, cows have been milked by milking machines rather than by hand for many decades [178]. In Brazil, a brand of dairy products uses a packaging design that includes the image of a peasant milking a cow by hand. A variety of products of this brand was present in all three Brazilian stores. There was no verbal message regarding the cow. The same kind of representation was also found in the UK involving an ice cream brand present at Asda and Tesco, where there also was no verbal message about the cow. We looked for further information online regarding the UK brand, and on the brand's website we found out, in a presentation video, that the system used was a milking robot. There is an anachronism in the way dairy product packages refer to dairy farming and the way it currently occurs. There are still representations of peasants, wooden buckets, and bell collars when the dairy industry has achieved high levels of automation. While the pastoral and owner's portrait modalities of representations attempt to revive a romanticised human–animal harmonic relationship, and all modalities claim some naturalness, an increasing number of farmers are adopting the milking robot system, which eliminates the need for human–animal interaction in the milking practice [178,183]. This is even a step ahead of mechanical milking. The use of the milking robot, as some argue, is less stressful for cows, but consumers tend to value more natural features, which have been showed to be a common bias amongst animal product consumers, similarly to what has been described by some authors about the power of the naturalistic fallacy on morality [39,184,185].

### 5.4. Appeal to Nature and the Misuse of Concepts from Animal Welfare Science

A range of studies show that consumers value naturalness in farm production [81,108,166,186–188], even when being informed that the free-range system can be worse than the zero grazing system in some aspects, such as increased aggressive behaviour, like pecking, or parasitic diseases in laying hens [81]. This suggests many people have a strong intuitive preference for naturalness, and it is not clear if there is a

bias to correlate naturalness and animal welfare regardless of the concrete facts. Indeed, this correlation seems reliable in many cases according to animal welfare guides and specialists [91], but this is not a principle or rule in animal welfare science. A question for further investigation is if consumers value animal welfare because they are concerned about animal suffering (attributing an intrinsic value to it) or if they value animal welfare because they believe it implies naturalness, often seen as an undisputed Good [184].

What is striking is how dairy marketing can take advantage of this bias to increase profit by forging associations between farming practice and naturalness [108,189]. Many seem to link a free access to the outdoors with naturalness [81,187,188]. So the grazing cow image on a package may assure the consumer of the naturalness of the milk despite the fact that mainstream milk production involves a range of biomedical interventions (e.g., antibiotic use), technological equipment, mechanisation, and concrete facilities [85,190,191].

Regarding meat, the increasing knowledge about animal emotions and cognition and the criticism about animal exploitation and intensive industrial production have triggered a new trend in marketing [107]. Perhaps this is reflected in dairy product marketing as well. Marketing communication has been opportunistically referring to animal welfare, once consumers have been shown to value it [35,78,192]. In doing so, marketing is misusing a technical term that has a specific meaning in the research area it comes from. According to Donald Broom [193], one of the pioneering scientists in this area, welfare is 'the state of an individual as regards its attempts to cope with its environment'; it is not something that can be provided but is a spectrum that varies from an extremely low to a very high level according to the success of coping with environmental challenges. When marketing communication misuses this expression, it is suggesting that welfare is an assured feature of that production system, and hence, of that product. In response to consumers' concerns about animal welfare, marketing is developing an animal welfare discourse and attempting to represent ideal animal welfare scenarios. However, the misleading messages and the way in which marketing is appropriating the animal welfare narrative has attracted criticism similarly to the controversial allegations about sustainability in eco and green marketing [108,194,195].

Based on our findings and related debates on critical animal studies about dairy marketing, we identified the following standard narrative in dairy product promotion: happy cow = healthy cow = better milk. This narrative incidentally reinforces the instrumental role of cows in the dairy industry, as their happiness is valuable only as a function of producing better milk.

In animal welfare science, the assessment of animal emotions and feelings is a meticulous and careful process to avoid misinterpretations and human bias as much as possible. Thus, to talk about animal welfare conditions accurately, a systematic investigation is indispensable, which might be based, for instance, on observations of species' specific behaviours, evaluation of hormone and neurotransmitter levels, and collection of blood, corporal fluids, or faeces samples [175,193]. The happy cow narrative alludes to happiness as an obvious and permanent feature of dairy cows. Supposing that the referred happiness is based on situations and states usually considered happy by humans, the referred happiness is naive and meaningless, or deceptive. In the cartoon modality, happiness is forged through unusual funny activities. Likewise, if the allusion to a cow's emotions on dairy packages is intended to be grounded in reality it would need to be attested and not be based on fallacies. Further, if it is based on a fictional narrative with no pretension of realism, this is totally useless for consumers' decision-making, and the risk of misleading people about the cows' lives is enough to make it objectionable.

In fact, some protocols to determine cows' emotions have been reported [196–198], but the happy cow allusion on dairy packages does not come from this. There is no study on cows' emotions stating that dairy cows are permanently happy, but there are plenty of studies indicating poor welfare conditions and listing welfare issues of dairy farming [84,88,89,199,200]. Recent studies on dairy cows' routine, welfare, and emotions indicate that cows under high productivity and commercial housing systems would hardly

be in a prolonged positive emotional state. But the dairy marketing, through the happy cow narrative, conveniently overlooks the plethora of stressing events that trigger negative emotions and, ultimately, cows' suffering [201–203]. The happy cow narrative oversimplifies and distorts complex subjects such as animal cognition, emotions, and sentience, and it is counter-productive to raise reflexivity and social responsibility about food choice implications on animal welfare.

Based on our findings and the current literature, we argue that many factors influence dairy attachment, as happens with meat attachment [204], and much of what is revealed by analysing meat-eating behaviour, such as the meat paradox, justifications, and moral disengagement, might encompass dairy consumption as well. In dairy product promotion and advertising, the natural/necessary/normal/nice (4N) attributes are widely present.

The happy cow narrative [112], also referred to as an idyllic rural setting [42,107–109,205] or dairy tale [107], might be an instance of moral disengagement, since it involves language manipulation that alleviates the weight of a harmful conduct and reduces personal responsibility [14,71,206]. Such a narrative was built under an ideology in which cows can and should be used as milk resources, and this precept determines cows' roles in society.

Our findings corroborate conclusions from other studies about the extremely positive depiction as well as the omission and lack of reliable information in dairy marketing regarding dairy cows' conditions [21,107,108,112,160,161].

*5.5. Limitations of the Study*

Some aspects that our study left uncovered were a systematic analysis of the content of packages with no explicit reference to cows and the differentiation of messages from products labelled as organic or free range. In addition, we did not perform an extensive examination of cultural differences, and this would be valuable for a better understanding and use of the results. Another aspect that would be worthwhile (for a more complex interpretation of the cows' meanings on packages) is a comparison of the history of the animal rights movement and tendencies in animal products marketing between both countries.

## 6. Conclusions

Our article adds to the findings mentioned in the broader literature that dairy product packaging relies upon dishonest modalities of representation which obfuscate the exploitation of nonhuman animals. Dairy product packages are probably the most relevant media between the dairy industry and consumers, as they are seen on the shelf during every shopping trip even by those who may not purchase the products. Therefore, the lack of relevant information about cows' exploitation—aspects that are actually related with their welfare—for products made with their milk and the deceptive claims about cows' lives should be included in the political agenda and critically addressed, taking into consideration the seriousness of this subject. Whether and how commercial media can reinforce an ideology is not obvious, and there is no consensus about it. Our study is not sufficient to address the impact of dairy package promotion on society's view of cows and is limited to indicate the incongruence between the social interest in global health, which includes cows' welfare per se, and the ethically controversial position of the dairy industry on cows' lives through the absent reference or deceptive reference to them in dairy product promotion. Furthermore, discussion and investigation about linguistic features like euphemistic labelling and positive messages associated with animal exploitation are needed to understand the impact of mass media communication on human–animal relations.

Additionally, further ethical discussion is essential to address whether dairy product promotion could meet global justice requirements regarding the negative impact of cows' milk production and the fact that it is not necessary for human health. Whilst counterarguments may stress that owing to food insecurity in parts of the world alternative nutrition sources may not presently by uniformly available, the existence of more nutritious and more sustainable alternatives to cows' milk poses to societies a moral requirement of discontinuing milk consumption, beginning in places where food infrastructure already

allows for this. Dairy alternatives already constitute a growing niche in many societies [207] and policy interventions could subsidise the most nutritious and sustainable of these. This would ameliorate the present situation of dishonest marketing and produce co-benefits for nonhuman animals, climate mitigation goals, biodiversity, and human health. In this sense, it is worth analysing what has been done in other cases in which the encouragement and promotion of certain products were considered harmful. Tobacco public policy around the world is a remarkable example and may serve to guide discussion on animal product regulation to limit marketing as well as public policies in order to decrease consumption. It is also valuable to learn from the current public efforts against the limitless advertising of foods that are high in sugar but low in nutritional content, especially targeting children. Despite their particularities, these examples can be useful because to some extent unhealthy foods, tobacco and animal-based foods have relevant similarities: their purchase is highly influenced by marketing, the industry interest in the increase of the consumption overlaps the need for behavioural changes that could improve quality of life of human and non-human animals, they have been criticized by an increasingly parcel of society. The companies behind these industries are generally large and they often claim that consumers have freedom of choice.

The decline of the dairy industry will likely impact rural workers, producers and families who base their economy on this activity and this needs to be carefully considered in public policies that will address this issue; but this contentious aspect is not exclusive to dairy production. There are other cases in which more sustainable measures can also impact the short-term workers and people who are economically dependent on a given sector, but their benefits in the medium- and long-term are beneficial for the society. In the case of dairy production, there are two important points to be prioritized: the first is that the subjugation and suffering of such many sentient individuals—the cows and the calves—who are forcibly implicated in the dairy industry, designates the dairy production as a huge ethical problem that demands urgent solution. The other point is that dairy food can be replaced by vegetable alternatives that also come from agriculture and can maintain or even generate more jobs and instigate innovation. A transition from dairy options to non-dairy would solve the ethical problem of subjugation and infliction of suffering over cows, could benefit human health, reduce diseases and impacts on the public health system, generate job opportunities and reduce environmental impacts of food production.

In summary, the overcoming of milk production is an urgent demand for sustainability, considering non anthropocentric definitions of the concept, which include the concern with animals' quality of life as a matter of social justice. The materialization of these benefits depends on development and planning, and it would not happen abruptly. In this sense, interrupting the indiscriminate encouragement of dairy consumption seems to be a first, minimal step, for which there is no justification for not taking.

**Supplementary Materials:** The following are available online at https://www.mdpi.com/article/10.3390/su13158418/s1.

**Author Contributions:** Conceptualization: K.C., C.P., R.T. and R.L.P.; data curation: K.C. and E.L.K.; formal analysis: K.C., C.P., R.T. and E.L.K.; funding acquisition: K.C. and R.L.P.; investigation: K.C., C.P. and R.T.; methodology: K.C., C.P. and R.T.; validation: K.C., C.P. and R.T.; visualization: K.C. and E.L.K.; writing—original draft: K.C.; writing—review and editing: K.C., C.P., R.T. and E.L.K.; supervision: C.P. and R.L.P. All authors have read and agreed to the published version of the manuscript.

**Funding:** This research was funded by CAPES foundation—Coordination for the Improvement of Higher Education Personnel.

**Institutional Review Board Statement:** Not applicable.

**Informed Consent Statement:** Not applicable.

**Data Availability Statement:** The data presented in this study are available in Supplementary Materials.

**Acknowledgments:** We thank the three anonymous reviewers who provided constructive comments and suggestions along the submission process. We also would like to thank Bruna Aparecida Bernardes da Silva, Gabriel Garmendia and Leo Heikiti Maeda Arruda for valuable input on the development of this manuscript and Adam Ford for the support during the data collection in the UK.

**Conflicts of Interest:** The authors declare no conflict of interest.

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
