# Peer review of "Exploring the Representation of Cows on Dairy Product Packaging in Brazil and the United Kingdom"

_sustainability, doi:10.3390/su13158418_

Round 1
Reviewer 1 Report
This paper provides a cogent analysis of dairy product packaging in the UK and Brazil, with the dominant theme being the absence/invisibility of cows themselves from that packaging, either completely, or through being substituted with more or less unrealistic representations. These findings concur with a large body of literature that has examined the self-serving misrepresentations of farming practices by industries that profit from nonhuman animal exploitation in diverse contexts. The comparison between the two countries is interesting, though I don't feel that the rationale for doing so is explained as clearly as it could be: Why were these two nations chosen (other than the fact of the researchers being located there)? What does this add to our understanding that wouldn't be gleaned from separate studies of the UK and Brazil, or from any other two countries? I'm sure a rationale can be provided, so it just needs to be much clearer - I'd suggest very early in the paper's introduction, there needs to be a clearer explanation of why this research team have collaborated on this project (anticipating, in some respects, some of the discussion at the start of section 2.1). One avenue to explore might be what can be learned in terms of nuancing local tactics to oppose the dairy industry more effectively, in different locations.
This leads on to a general area where the paper could be strengthened, which is to clarify the substantive sections, perhaps with some restructuring: Firstly, the introduction is extremely long - it reads more like a literature review than an introduction, so I recommend a new, shorter, intro, which explains the background to the research (i.e. why this team, why this comparison, what the research question and aims were) and signposts the structure of the rest of the paper. Then most of what's in the current intro could be re-framed as a literature review/background to the study (and could probably be much more concise). Within the existing intro, there seems to be an imbalance in the coverage of the three 'main axes' (p.3) - i.e. human health issues get very little mention - if flagging up 3 main axes, I'd expect to see them get roughly equal coverage in the paper. Section 1.2 as a whole seems to switch around between axes, rather than provide an overview of each in turn.
Secondly, the methods section - there isn't much discussion of why the sample was narrowed down as it was, just a statement that it happened - what was the rationale for this? (p.7, para 2). Likewise, the decision to 'focus on the three largest chains in each country' - why? Easy to justify perhaps, but it needs to actually be done within the paper. And if doing so, the market share of each chain needs to be provided. I also wondered about any disjuncture between nation and locality - the 3 largest chains of each country have been chosen, but the sample is obtained from 2 specific cities - are these also the largest chains in these cities, or not? If not, this needs to be accounted for - if so, make sure to make that point that the localities are congruent with the national picture (otherwise, why make the point about national market share if not conducting a national study?) More generally, there is very little discussion of methodology, i.e. how does this study relate to visual methodologies more generally? Why choose the three categories of icon, index and symbol? How does this relate to the methodology used by other critical analyses of dairy/animal product marketing? Does it build on them, offer something novel to them, improve on them in some way, etc.?
Thirdly, the results section: it's striking that there's relatively little discussion of the majority of the packaging that lacks cow representations (given that this is flagged in the abstract). This begs the question of what's on the packaging instead of cow representations? And what's the significance of this? Some commentary about this in terms of the absent referent seems appropriate, as a minimum (in the discussion).
Finally the discussion section - this is lengthy but not broken down into subsections - some breaking up with headings could be useful, e.g. relating to key themes from the study. Given the (appropriate) use of the absent referent to analyse the data, I think Adams' work needs to be foregrounded in the paper's introduction and abstract. In the discussion, key literature could be drawn on, specifically Cole and Stewart's work on childhood socialisation (e.g. in their 2014 book Our Children and Other Animals, and elsewhere) - given that the paper does highlight the targeting of children as consumers, this seems to be an essential reference point for the analysis, not least because C&S specifically address the use of unrealistic representations of nonhuman animals, including as 'cute' cartoon representations. It also provides an analytical framework for understanding this lack of visibility of exploited nonhuman animals, as well as the denial/minimisation of their subjectivity, both of which feature prominently in the discussion. Given the specific topic of this paper, another valuable reference point would be Cole's 2017 chapter 'Hiding and legitimating oppression in dairy product advertising' (in Nibert, D. (ed) Animal Oppression and Capitalism, vol.1).
Some more minor points:
The reference to 'moral schizophrenia' on p.2 is problematic - Francione's use of this term has been rightly criticized as ableist, given that it deploys mental illness as a metaphor in such a way as to reproduce harmful stereotypes about schizophrenia. This reference should be removed/replaced with an alternative - or simply spend longer explaining the 'meat paradox' that comes later in that same paragraph.
The authors advance cows' interests in not being exploited in the paper, and on that basis, I don't think it's defensible to use the industry euphemism of 'livestock' in the paper, without at least putting the term in scare quotes - this occurs on p.3 at the start of section 1.2, and in the final para on p. 11, for instance.
On p.4, the paper makes claims about projected increases in milk consumption, but the reference here is dated - 2009 - are there more up to date projections than this? If not, I think it's important to add a sentence of caution as to how relevant this projection still is, given that the dairy industry (and animal-industrial complex in general) is under sustained and increasing critique for its role in global crises, some of which this paper comments on itself.
On p.5, the break down of types of textual messages could be explained more clearly - i.e. secondary, romance (especially this one), mandatory - where do these terms come from?
p.9 - the phrase 'head upside toward the grass' - can this be rephrased? I couldn't understand what this meant. Also on p.9 'bubble speech' should be 'speech bubble'.
p.11 - the sentence that includes 'dairy fairy tale' is long and hard to follow, can this be reworded/broken up?
The conclusion: I felt this could be much stronger, i.e. making a clearer critical intervention on behalf of cows - no 'further ethical discussion' is required to conclude that the dairy industry is morally abhorrent and cannot 'meet global justice requirements', even if it were not also environmentally problematic, etc. Instead, the focus should be squarely on how to dismantle the dairy industry and replace it (and the entire animal-industrial complex) with a rational, equitable, sustainable food system(s). In short, I feel a much stronger ethical, critical statement is needed to conclude the paper - the basic argument appears to be that the industry is lying to consumers, including cynically targeting children, in ways that are massively harmful to consumers, the environment and to cows themselves, but that gets a little lost in the conclusion.
Author Response
Thank you very much for your very useful comments and suggestions. Unfortunatelly I couldn't meet all your requirements as I wish due the limited time.

Reviewer 2 Report
Within the tradition of critical animal studies, the submitted paper aims to explore the representations of cows on dairy products packaging at two countries: UK and Brazil. This is a well written paper, with the potential of becoming a significant contribution to the field. However, some issues require substantial improvement.
It is clear that the authors defend a socialist and animal-rightist standpoint. In the introduction, the authors condemn capitalism and economic powers lato sensu and advocate for the abolition of animal use. The authors, who are established experts in their field, are obviously entitled of voicing their opinions on such contentious issues. However, by punctuating the introduction with ideologically-laden statements, such as “This illustrates the constant conflict of interest between corporate capitalism and social goods” and “replacing them seems to be one of the steps that need to be taken as dairy promotion becomes more and more unjustifiable”, the objectivity of the research is undermined. Given the empirical nature of the paper, I would recommend the authors to abstain from these considerations, that may be construed as a cognitive bias.
The introduction is very informative but arguably excessively long, and some aspects could be moved to the discussion. I recommend that the aims of the paper should be more clearly described towards the end of the introduction.
The description of the methods requires substantial improvement since several questions remain unanswered:
- Why were these two countries selected?
- Why were not cultural differences explored?
- What evidence exist that these retailers are “the main supermarket chains”? Why not Sainsbury’s or Zona Sul? This seems more a convenience sample than a purposeful sample.
- When was the data collected?
- Did you buy all these products or just took pictures of them? Did you take pictures of the full package or just the front?
- In terms of imagery, how have you reached the four modalities? You say: “We distinguished and named modalities from (Karnal, Machiels, Orth, & Mai, 2016) representation that we found in the preliminary survey.” This paper does not contain any of the modalities that you use. And to which preliminary survey do you refer?
- Who performed the analysis?
- Was the analysis validated? Were any measures in place to avoid anedoctalism?
- How were the verbal messages analysed? What were you looking for?
Regarding Results, some concerns arise. These results cannot be replicated because the reader ignores which dairy products were analysed. Please provide a detailed account of the analysed products as appendix.
“Regarding the graphic design patterns, we only counted each design once”. This segment belongs to the methods. In addition, how have you dealt with ambiguous patterns?
Description of results from the verbal message is insufficient. It is unclear which information was retrieved. What is meant by bubble-speech? The discussion explores some of these findings, but a more systematic description is warranted.
The perception of a potential cognitive bias (that the researchers are trying to prove that they are right rather than to, dispassionately and objectively, address the research topic) is reinforced when reading the discussion. You conclude that: “cows appear in milk advertisement as instruments of production (Lerner & Kalof, 1999); our findings corroborate this, especially in the cartoon and pastoral categories.” I beg to differ. First, because your study shows that roughly 68% of dairy packages in UK and 85% of those in Brazil, do not represent images of cows altogether. This finding needs to be described and discussed (including possible explanations for these differences). Secondly, you seem to imply that the pastoral design is intrinsically misleading or even false. But that is hardly self-evident, for several reasons. One possible reason is that your assessment does not consider the production system (intensive, semi-intensive, or organic). The other is that dairy farming is much more nuanced than described in the paper. Take Kerrygold butter, for example. Most cows in Ireland have access to pasture and are grass-fed (at least for the most part of the year), so their labelling cannot be considered deceptive. And why can’t a realistic photograph of a grassland with dairy cows denote a pastoral modality? Finally, the idea that dairy cows do not experience happiness is as misleading as describing them as happy cows. And yes, there are studies on cows’ emotions confirming that dairy cows can experience positive emotional states (https://www.nature.com/articles/s41598-021-84371-x).
Another argument that I found misleading is the comparison between milk and tabacco. Tabacco has no significant role in society. You can suppress it without being replaced by anything else. Dairy products, however, are highly nutritious and a substantial part of western diets and cannot be suppressed without being replaced by suitable alternatives. Any meaningful discussion regarding this issue needs to consider the alternatives to dairy products.
A section identifying the limitations of the study is currently absent. Please add.
The paper is well grounded on relevant literature. Please check your references because some typos were detected (Refs. 14, 19, 103, 104 – The name of the journal is missing, Ref. 166 – Delete Richard Twine; Ref. 172 – One author and the name of the journal are missing). Bastian & Loughnan, 2017 and Onwezen & van der Weele, 2016 do not explore the role of livestock as a significant cause of environmental damage (p.3).
Author Response

(The authors gave the same response as above.)

Reviewer 3 Report
The reviewed manuscript addresses a really fascinating research question, which links marketing, food production, and agriculture, and put these into the frame of sustainability via attention to the animal exploitation issue. The article is very informative. It is strong both theoretically and empirically. It is also well-written and well-structured, and it bears a lengthy list of suitable references. I'm sure it will attract significant attention of the international research community. After several rounds of the critical reading, I see only three minor issues for improvement.
- What is now Introduction can be labeled as 'Conceptual Background' to follow a much shorter Introduction. The latter should briefly outline the problem and its international urgency and pose the objective of this study,
- Can you provide some elementary statistics about dairy product consumption in the considered countries?
- Can you add any figure (may be with the noted statistics)? One figure for so lengthy paper is not enough. Moreover, several figures would increase readability of your work.
Author Response

(The authors gave the same response as above.)

Round 2
Reviewer 1 Report
The revised paper has added clarity thanks to the revised introduction and has successfully addressed many of the more minor points from the first review. However, there has been no attempt to engage with the additional literature - I don't think it's acceptable to cite a lack of time as a reason not to read, use and cite literature that is so closely related to the topic of the paper. So, I can only reiterate points 7 and 14 in particular from my initial review and ask that those revisions be made to the paper. In addition, some of the revisions need careful checking for the use of English.
Author Response
Thank you for your comments and suggestions. Please see attached our reply.

Reviewer 2 Report
The authors have accommodated some suggestions that greatly improved the paper but some issues remain unanswered. For example, the description of the analysis has remained unchanged:
- In terms of imagery, how have you reached the four modalities? You say: “We distinguished and named modalities from (Karnal, et al. 2016) representation that we found in the preliminary survey.” This paper does not contain any of the modalities that you use. And to which preliminary survey do you refer?
- Who performed the analysis? You need to differentiate, within the research team, who did what.
- How were the verbal messages analysed? You need to explain how content analysis was performed. Simply saying “We only analysed the verbal message that was referring to cows” is not enough.
- An appendix should be included, disclosing the products that were analysed at each store. Only this can ensure the validity of your analysis (given the fact that no external validity measures were in place).
The English level of the revised segments is not at the same standard of the rest of the paper. Please reside. Also, the authors, have failed to address some of the issues identified by both reviewers 1 and 3, on the grounds of having very limited time. This is not an acceptable justification. Please respond to these comments.
As per Manuscript Preparation Guidelines, limitations of the study must be part of the discussion section and not a section separated from the main paper. Please revise.
Additional comments can be found an annex.

Author Response

(The authors gave the same response as above.)

Round 3
Reviewer 1 Report
The revised paper retains the strengths of the original submission, but it is now much improved overall: the grounding in relevant literature is stronger, the structure is clearer and the conclusion is much stronger and more effective. The latter in particular I think will make the paper more useful for future researchers to cite and build on, and the section on 'limitations' is likewise helpful for stimulating further research (perhaps including by the current authors).
The text is also much improved in terms of the use of English language, but there are still some minor corrections needed - my review copy in this round didn't have line numbers, but for instance on p.2: "since fruits and vegetables intake are bellow recommended" - correct 'bellow' to 'below' and add 'levels' after 'recommended'. In the next paragraph on p.2 'shelfs' should be 'shelves'. So, a thorough proof-read is still needed to finalise the paper for publication.
But overall, the paper in its revised form makes a valuable contribution to the field and I look forward to reading the final published version in due course.
Reviewer 2 Report
The paper has very much improved, and I only have some minor suggestions, to correct errors and remove repetitions, and that will hopefully improve its readability.
Section 2.2. – “In this article we will give more attention to the animals’ interests in not encouraging animal products consumption, considering it a sufficient reason while the environmental damage is an aggravating factor, that poses an existential risk not only for humans, but for biodiversity generally.” This sentence is complex and poorly written. Please rephrase!
You now have reference to the selection of supermarket chains at two different places in the methods section. This needs tidying up. I suggest the following:
Start Section 3.1 with: “At an initial stage, during the first half of 2017, we visited a range of popular retail stores in both the UK and Brazil, taking pictures of animal representations from diverse categories of animal products.
In section 3.1, move the section “Three Chains…. and higher prices” to a new section 3.2 - Case Selection
Section 3.1, last paragraph - “three retail chains” instead of “the three largest chains”.
New Section 3.2 – Case selection (or maybe Retailer selection)
Three chains from each country were chosen due to their wide distribution throughout the national territory and variety of products (from several regions of the country, reflecting a good national sample) that target different classes. In Brazil, data was collected in Rio de Janeiro, the second most populous city of the country, from three leading supermarkets: Extra – from the Companhia Brasileira de Dis-tribuição, Carrefour, and Prezunic – from the Cencosud company. The chain Zona Sul or the Pão de Açúcar stores focus on upper economical classes and display less variety and were thus not included in the study. Also, these are not well spread over the cities: Zona Sul is only present in the south and central area of Rio de Janeiro, while Extra, Carrefour and Prezunic are widely spread and target middle and lower classes. Similar counterparts were selected in the UK: Tesco, Asda and Aldi also target middle and lower classes, offer a big variety and are well spread even in marginal areas over the cities, while Sainsbury’s chain, on the other hand, has smaller stores, less variety of products and higher prices. We visited the biggest store of each of the three chains in the Liverpool City region, the fifth largest metropolitan area in the UK.
Section 3.3 Data Collection
Start this section with “The sampling procedure consisted of taking photos of the products on the shelves in each visited store.”
Remove “In both the UK and Brazil…” because it was moved to section 3.2
Move “Data collection occurred between 2017 and 2018” to the end of this section.
3.4 Imagery Selection
Delete “To analyse the package imagery, we took photographs for the analysis.” (Redundant)
Discussion –
“around 85% of the packages from Brazil and 70% of the packages from UK” instead of “around 70% of the packages from Brazil and 85% of the packages from UK”
